# Evolution of foraging behaviour induces variable complexity-stability relationships in mutualist-exploiter-predator communities

Lin Wang[1,2,3], Ting Wang[4], Xiao-Wei Zhang[5], Xiao-Fen Lin[6], Jia Li[2], Jin-Bao Liao[7]*, Rui-Wu Wang[8]*

1 Ministry of Education's Key Laboratory of Poyang Lake Wetland and Watershed Research, Jiangxi Normal University, Nanchang, China, 2 School of Geography and Environment, Jiangxi Normal University, Nanchang, China, 3 Jiangxi Key Laboratory for Intelligent Monitoring and Integrated Restoration of Watershed Ecosystem, Jiangxi Normal University, Nanchang, China, 4 Hunan Institute of Agricultural Economics and Information, Hunan Academy of Agricultural Sciences, Changsha, Hunan, China, 5 College of Life Sciences, Shaanxi Normal University, Xi'an, China, 6 School of Mathematics and Statistics, Northwestern Polytechnical University, Xi'an, China, 7 Ministry of Education Key Laboratory for Transboundary Ecosecurity of Southwest China, Yunnan Key Laboratory of Plant Reproductive Adaptation and Evolutionary Ecology and Centre for Invasion Biology, Institute of Biodiversity, School of Ecology and Environmental Science, Yunnan University, Kunming, China, 8 College of Life Sciences, Zhejiang University, Hangzhou, China

* jinbaoliao@ynu.edu.cn (JBL); wangrw@nwpu.edu.cn (RWW)

## Abstract

Early ecological theory predicts that complex ecological networks are unstable and are unlikely to persist, despite many empirical studies of such complexity in nature. This inconsistency has fascinated ecologists for decades. To resolve the complexity-stability debate, coupling population dynamics and trait dynamics is considered to be an important way to understand the long-term stability of ecological community assemblages. However, we still do not know how eco-evolutionary feedbacks affect the relationship between complexity and stability in ecologically realistic networks with both antagonistic and mutualistic interactions. Here, we explored an adaptive network model to evaluate how the evolution of foraging preference to determine the relationship between network complexity (i.e., connectance) and stability (i.e., community persistence at steady state) in mutualist-exploiter-predator communities (MEST). Our theoretical results showed: (i) adaptive foraging of the top predator contributes to the stability of mutualism and intermediate intensity of foraging adaptations can lead to chaotic dynamics in a four-species MEST community; (ii) the complexity-stability relationship may show positive monotonic, negative monotonic, peaked and double-peaked patterns in general MEST communities, while the double-peaked pattern is only obtained when both the adaptation intensity and interspecific competition are high. Furthermore, model predictions may be consistent with both the negative monotonic pattern revealed in freshwater communities and the peaked pattern revealed in marine communities. Finally, we infer that foraging

**Data availability statement:** No data were collected for this study and simulation code is available at https://github.com/lwang191206/Adaptive-food-web-model.

**Funding:** L.W. acknowledges the support of both National Natural Science Foundation of China (32201259) and Natural Science Foundation of Jiangxi Province (20224BAB215010); X.-W.Z. acknowledges the support of the National Natural Science Foundation of China (32201264); J.-B.L. acknowledges the support of the National Natural Science Foundation of China (32271548) and R.-W.W. acknowledges the support of National Natural Science Foundation of China-Yunnan Joint Fund (U2102221). The funders had no role in the study design, data collection and analysis, decision to publish, or preparation of the manuscript.

**Competing interests:** The authors have declared that no competing interests exist.

adaptations of the top predator may alter positive or/and negative feedback loops (trait-mediated indirect effects) to affect the stability of general MEST communities. Our adaptive network framework may provide an effective way to address the complexity-stability debate in real ecosystems.

## Author summary

In our recent work, we investigated the intricate relationship between network complexity and stability within diverse communities that exhibit both mutualistic and antagonistic interactions. Traditional ecological theories often suggest that complex networks are inherently unstable, yet nature is full of such complexity. To bridge this gap, we developed an adaptive network model that considers both mutualist-exploiter-predator dynamics and the evolution of foraging preferences. Our findings reveal that adaptive foraging strategies can significantly enhance the stability of mutualistic interactions while also introducing potential chaos under certain conditions. We identified various complexity-stability relationship patterns, indicating that both the intensity of foraging adaptations and interspecific competition play critical roles. This insight could help explain the observed variability in stability across different ecological communities. Ultimately, our adaptive network framework offers valuable insights that may help resolve ongoing debates about the stability of complex ecological networks, benefiting both scientific communities and conservation efforts aimed at preserving biodiversity.

## Introduction

May's local stability analysis of randomly connected species in food web models showed that increasing network complexity (e.g., connectance) leads to decreasing stability [1], but this result is contradictory to earlier empirical findings [2,3]. Due to the inconsistency, research of complexity–stability relations became one of the most challenging issues in theoretical ecology [4,5]. In fact, May's model is the random matrix, while real ecosystems are not assembled purely at random [6]. The challenge of understanding the structures and interactions in real ecological networks that deviate from May's assumptions remains a difficult problem, and it has been a central focus of food web ecology for decades [7–9].

  Previous research revealed three possible relationship patterns between connectance (e.g., the ratio of actual links to potential links), one of important metrics of the network complexity [10], and stability (e.g., locally stable if $Re(\lambda_{max})<0$) in model and empirical ecosystems: negative correlation [1], positive correlation [11] and no correlation [8]. For instance, early theoretical studies found that increasing the connectance will decrease feasibility (i.e., positive equilibrium solution) [12] and global asymptotic stability [13], although indices of the complexity and stability in these studies vary; ecologists asserted that complex food webs are rarely likely

to persist due to the inherently unstable properties of complex systems. On the contrary, by excluding unrealistic food web structures, Plitzko et al. [14] revealed the existence of positive relation between network connectance and stability within realistic parameter regions. In addition, by conducting a stability analysis of 116 quantitative food webs sampled worldwide, Jacquet et al. [8] found that complexity (connectance) is not associated with stability and speculated that empirical ecosystems have several non-random properties (e.g., interaction strength topology and frequency distribution) contributing to the absence of a complexity–stability relationship. However, there is still a lack of theory to reconcile the complexity-stability debate in more realistic network structures.

One potential reason for these variable complexity-stability relationships could be changes in the dimensionality of the system (i.e., ecosystem size and interaction types). Recent studies have found that the increase in the dimensionality can change the cases from stable to unstable states. Main examples include the omnivory-stability relationship [15], the diversity-stability relationship [16] and the link-strength variability-stability relationship [17]. In fact, incorporating pheno-typic trait dynamics into the single ecological framework can not only greatly increase the system dimension, but may change community stability [18–20]. However, we still do not know how eco-evolutionary feedbacks affect the relationship between complexity and stability in ecologically realistic networks.

Adaptive foraging [21], an important behaviour trait, and is thought to potentially affect the stability of food webs [22,23] and mutualistic networks [24,25]. For instance, Kondoh [22] added adaptive foraging (AF) into a food-web model with linear nonsaturating functional responses, and found that complexity (connectance) may enhance community stability (i.e., commu-nity persistence) through the facilitation of dynamical food-web reconstruction; without the AF, food-web complexity reduces the persistence. In addition, Uchida and Drossel [23] found that foraging behaviour has a large stabilizing effect, which leads to a positive complexity–stability relationship in food webs. In mutualistic networks, Valdovinos et al. [24] found that the AF reverses positive effects of connectance on the community stability by partitioning the niches among species within guilds. Overall, compared without the AF, the above findings seem to suggest that the AF may reverse the complexity-stability relationship in food webs or mutualistic networks. In fact, real ecological communities often contain both antagonistic and mutualistic relationships [26–28]. For instance, by integrating pollinators into food webs, Hale et al. [26] found that the mutu-alisms can increase the persistence and temporal stability in multiplex networks. However, little is known about community dynamics driven by the AF in real ecosystems with both mutualistic and antagonistic interactions.

Recently, we have developed a theoretical framework that incorporates both mutualistic and antagonistic relationships [29]. The model framework includes four species across three trophic levels: top predator, specialist predator, mutualist and exploiter. First, the exploiter competes with the mutualist for resources and both of them are preyed upon by the top predator, and the mutualist-exploiter-top predator interaction is similar to apparent competition [30]. Moreover, the mutualist-specialist predator-top predator interaction is usually treated as omnivory [31]: the specialist predator only feeds on the mutualist, while the top predator feeds on both the specialist, mutualist and exploiter. Significantly, because the generation span of the fig tree is much longer than both fig wasps (i.e., specialist predator, mutualist and exploiter) and the ant (i.e., top predator), this theoretical framework ignored the basal resource (i.e., fig tree) and treated the mutualist as one potential resource [29]; meanwhile, it assumed that foraging preferences of the top predator to all prey were fixed constants. In fact, a flexible network structure driven by foraging preferences of predators often vary in time and space [32,33]. However, it is still unclear how adaptive foraging of the top predator affects the complexity-stability relationship in mutualist-exploiter-specialist predator-top predator (MEST) communities.

In what follows, we establish an adaptive network model to evaluate the effects of the adaptive foraging on the relation-ship between network complexity (i.e., connectance) and stability (i.e., community persistence at steady state) in MEST communities. This study mainly focuses on answering two key scientific questions: (i) what are the intrinsic mechanisms by which variable complexity-stability relationship emerges in an adaptive dynamic framework and (ii) how to reconcile the complexity-stability debate in model and empirical ecosystems. Our work may offer valuable insights on the stability of complex ecological networks.

## Methods

### Study system

To investigate the complexity-stability relationships in ecological communities, this work extends a recent study (Fig 1A) of omnivorous food-web framework [29]. First, the community includes a variety of exploiters. For instance, up to 32 species of non-pollinating wasps (competitors of the pollinating wasp) can be found in *Ficus microcarpa* [34]. Second, unlike the previous model [29], dietary preferences of the top predator to prey are not fixed, but change over time. For instance, early studies found that foraging strategies of the ant are characterised by flexibility [35] and adaptability [36]. On this basis, we can obtain a new research framework (Fig 1B). In Fig 1B, as the top predator ($P$) flexibly adjusts its foraging strategies according to feeding preferences (i.e., the width of solid lines represents the intensity of feeding preferences of $P$ to all prey). Significantly, because the generation span of the basal resource (i.e., fig tree) is much longer than other species in Fig 1A, our adaptive network framework ignores the basal resource (i.e., fig tree) and treats the mutualist as one potential resource.

### Adaptive network model

Recent empirical work found that feeding rates may commonly be unsaturated (i.e., Holling type I functional response or linear function) and that the degree of saturation varies with a variety of factors including body size, habitat, interaction dimension and temperature [38]. In this study, based on the network framework presented in Fig 1B, we established an adaptive network model with the Holling type I functional response:

$$\frac{dF_0}{dt} = F_0 \left( \underbrace{r_0}_{\substack{growth \\ rate}} - \underbrace{\alpha_0 F_0}_{\substack{density- \\ dependent \\ death}} - \underbrace{\sum_{i=1}^{n} \beta_{i0} F_i}_{\substack{interspecific \\ competition}} - \underbrace{a_C C}_{\substack{foraged \\ by\ C}} - \underbrace{\theta_0 u_0 P}_{\substack{foraged \\ by\ P}} - \underbrace{d_0}_{\substack{mortality \\ rate}} \right),$$

(1)

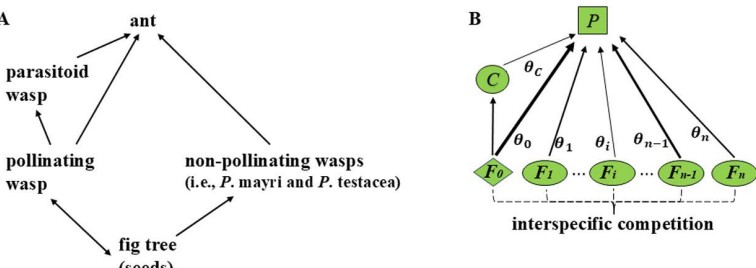

**Fig 1. A graph of relationships among top predator (*P*: ant), specialist predator (*C*: parasitoid wasp), mutualist (*F0*: pollinating wasp) and exploiters (*Fi* (i = 1, ..., n): non-pollinating wasps).** (A) empirical ecosystem work [37] and (B) an adaptive network framework in mutualist-exploiter-predator communities. $\theta_j$ is the foraging effort of $P$ to $F_j$ ($j = 0, 1, \ldots, n$), $\theta_C$ is the foraging effort of $P$ to $C$ and $\theta_C = 1 - \sum_{j=0}^{n} \theta_j$. Solid lines indicate predation and each exploiter competes with the mutualist and other exploiters for resources generated by the mutualist; the width of solid lines represents the intensity of selective preferences of $P$ to all prey.

$$\frac{dF_i}{dt} = \underbrace{r_i F_i \left(1 - \frac{F_i}{\frac{F_0}{q}}\right)}_{\substack{\text{mutualism–}\\\text{dependent}\\\text{growth}}} - \underbrace{\sum_{j=0,\ j\neq i}^{n} \beta_{ji} F_j F_i}_{\substack{\text{interspecific}\\\text{competition}}} - \underbrace{\theta_i u_i F_i P}_{\substack{\text{foraged}\\\text{by } P}} - \underbrace{d_i F_i}_{\text{death}}, i = 1, 2, \ldots, n,$$

(2)

$$\frac{dC}{dt} = C \left[ \underbrace{e_C a_C F_0}_{\substack{\text{growth caused}\\\text{by consumption}\\\text{of } F_0}} - \underbrace{\left(1 - \sum_{k=0}^{n} \theta_k\right) u_C P}_{\substack{\text{foraged}\\\text{by } P}} - \underbrace{d_C}_{\substack{\text{mortality}\\\text{rate}}} - \underbrace{\alpha_C C}_{\substack{\text{density–}\\\text{dependent}\\\text{death}}} \right],$$

(3)

$$\frac{dP}{dt} = P \left[ \underbrace{e_P \sum_{k=0}^{n} \theta_k u_k F_k}_{\substack{\text{growth caused}\\\text{by consumption}\\\text{of all exploiters}}} + \underbrace{e_P \left(1 - \sum_{k=0}^{n} \theta_k\right) u_C C}_{\substack{\text{growth caused}\\\text{by consumption}\\\text{of } C}} - \underbrace{d_P}_{\substack{\text{mortality}\\\text{rate}}} - \underbrace{\alpha_P P}_{\substack{\text{density–}\\\text{dependent}\\\text{death}}} \right] \equiv P W_P,$$

(4)

$$\frac{d\theta_j}{dt} = g\theta_j \left( \underbrace{\frac{\partial W_P}{\partial \theta_j}}_{\substack{\text{fitness}\\\text{gradient}}} - \underbrace{\sum_{k \in sp.\ P's\ prey\ resources} \theta_k \frac{\partial W_P}{\partial \theta_k}}_{\substack{\text{mean fitness}\\\text{gradient}}} \right), \ , \ j = 0, 1, 2, \ldots, n.$$

(5)

the variable $F_0$ is the mutualist biomass, $F_i$ ($i = 1, 2, \ldots, n$) is the exploiter biomass, $C$ is the biomass of the specialist predator, $P$ is the biomass of the top predator, $\theta_j$ is the foraging effort of $P$ to $F_j$ ($j = 0, 1, 2, \ldots, n$), and function $W_P$ defines the fitness [39] of $P$. Model parameters: $r_0$ is the growth rate of $F_0$, $r_i$ is the growth rate of $F_i$; $\alpha_0$, $\alpha_C$, $\alpha_P$ are density-dependence coefficients of species $F_0$, $C$ and $P$, respectively; $\beta_{i0}$ is the interspecific competition coefficient of $F_i$ to $F_0$, $\beta_{ji}$ is the interspecific competition coefficient of $F_j$ to $F_i$; $a_C$ is the consumption rate of $C$ to $F_0$, and $u_0$, $u_i$, $u_C$ are consumption rates of $P$ to $F_0$, $F_i$ and $C$, respectively; $e_C$ is the conversion efficiency of $C$, $e_P$ is the conversion efficiency of $P$; $d_0$, $d_i$, $d_C$ and $d_P$ are mortality rates of $F_0$, $F_i$, $C$ and $P$, respectively.

In Eq. (2), similar to early viewpoint that carrying capacity as a function of density of interaction partners [40,41], $\frac{F_0}{q}$ acts as the capacity of $F_i$ and $q$ is a scale coefficient [29]. The parameter $q$ presents the strength of the system's dependence on mutualistic relationships, while the new framework presented in Fig 1B will become the food web model without

mutualistic interactions when $q \to 0$; moreover, when $q > 0$, all exploiters are dependent upon the mutualist's facilitation of its resource. Specifically, since the life cycle of fig trees is much longer than that of fig wasps (e.g., the pollinating fig wasp $F_0$, non-pollinating fig wasp $F_i$, and parasitoid wasp $C$) [42,43], we set the mutualist ($F_0$) to represent the mutualistic relationship between the fig tree and its pollinating fig wasp, and the development of $F_i$ depending on the mutualist (i.e., $F_0/q$). In Eq. (5), the evolution of foraging preference ($\theta_j$) derives from Lagrange-multiplier methods [44,45] and $g$ is the intensity of foraging adaptation.

## Model analysis

In the adaptive network model, all model parameters (Table 1) can be chosen based on the stability criteria [46] under the non-negative equilibrium. We calculate a positive equilibrium of species coexistence and analyze the maximal real part of all eigenvalues (i.e., $Re(\lambda_{max})$) under the equilibrium, and obtain a locally stable state when $Re(\lambda_{max}) < 0$. Similar to our early theoretical work in the fig-wasp system [29,47], the unit of each parameter is dimensionless.

In the mutualist-exploiter-specialist predator-top predator (i.e., four-species MEST) community, first, we fixed other parameters and analyzed how the intensity ($g$) of adaptive foraging affects community stability and population dynamics (e.g., periodic and aperiodic/chaotic dynamics; S1 Appendix). Second, because the exploiter ($F_1$) is a disruptor of the mutualistic relationship and is subject to interspecific competitive pressure ($\beta$) from the mutualist ($F_0$) and predation pressure ($u_1$) from the top predator ($P$), we fixed the adaptation intensity ($g$) and analyzed the effects of key parameters (i.e., $\beta$, $u_1$) changing on food web structure and stability (i.e., calculating the $Re(\lambda_{max})$ at the equilibrium point of the four-species MEST model based on the analytical solution method [29]). Finally, we do sensitivity analyses of model parameters (S2 Appendix) and model structures (S3 Appendix).

Notably, when network size is small, we can easily list various network structures (e.g., each boundary equilibrium point of the adaptive network model) and calculate the $Re(\lambda_{max})$ at the equilibrium point separately; however, when the network size is large (i.e., network size is set to 35), it is more difficult to list all possible network structures (e.g., in the 35-species community, the network structure will change when the top predator selectively preys on the specialist predator and 32 exploiters, so the total number of boundary equilibrium points is $2^{33} - 2$) and calculate the $Re(\lambda_{max})$.

Therefore, in general MEST communities (e.g., network sizes are set 5, 10, 15, 20, 25, 30, 35), we used community persistence (the fraction of remaining species in the system after running the simulations long enough that this fraction stayed constant) as a measure of the stability in complex systems. Then, we investigated how interspecific competition ($\beta$), foraging adaptation ($g$) and network size ($N$) affect the relationship between system stability (e.g., community persistence) and network connectance. The connectance can be calculated by the ratio of actual links (i.e., connection between the top predator and its prey) to potential links (i.e., fully connection); when the connectance is less than 1, some

**Table 1. Parameters used in the adaptive network model.**

| Par. | Description | Value | Par. | Description | Value |
|---|---|---|---|---|---|
| $r_0$ | growth rate of $F_0$ | 0.5 | $\alpha_0$ | density-dependent coefficient | 0.13 |
| $\beta_{ji}$ | interspecific competition of $F_j$ to $F_i$ | $\beta_{ji} = \beta \in [0,\ 0.18]$ | $a_C$ | consumption rate of $C$ to $F_0$ | 0.2 |
| $u_0$ | consumption rate of $P$ to $F_0$ | 0.11 | $d_0$ | mortality rate of $F_0$ | 0.05 |
| $r_i$ | growth rate of $F_i$ | [0.3, 0.4] | $q$ | scale coefficient | 0.25 |
| $u_i$ | consumption rate of $P$ to $F_i$ | [0, 0.3] | $d_i$ | mortality rate of $F_i$ | 0.05 |
| $e_C$ | conversion efficiency of $C$ | 1 | $u_C$ | consumption rate of $P$ to $C$ | 0.2 |
| $d_C$ | mortality rate of $C$ | 0.05 | $\alpha_C$ | density-dependent coefficient | 0.12 |
| $e_P$ | conversion efficiency of $P$ | 1 | $d_P$ | mortality rate of $P$ | 0.05 |
| $\alpha_P$ | density-dependent coefficient | 0.1 | $g$ | adaptation intensity | [0.05, 0.5] |

of the exploiters may not be foraged by the top predator and rely on the commons for resources (e.g., seeds produced by the pollination of the pollinating fig wasp) generated by the mutualist $F_0$ [29]. Moreover, we explored the potential for species coexistence in a complex MEST community (S4 Appendix). Furthermore, we combined theoretical predictions with empirical data analyses (S5 Appendix). Finally, we offered a potential mechanism for the foraging adaptation affecting community stability. In model simulations (via ODE45 in Matlab R2016a), species and foraging efforts were deemed extinct if the biomass fell below $10^{-12}$.

## Results

### Foraging adaptation and stability in a simple system

In the mutualist-exploiter-specialist predator-top predator (i.e., four-species MEST) community, our adaptive network model exhibits a complex dynamic behaviour as the intensity ($g$) of foraging adaptations increases (Fig 2).

As $g$ increases, the population dynamics of the exploiter ($F_1$) and preference trait ($\theta_1$) dynamics evolve from simple periodic behaviour ($g=0.08$, case I; Fig 2A, 2C and 2D), period-doubling oscillations ($g=0.22$, case II; Fig 2A, C, D) into chaos ($g=0.28$, case III; Fig 2A, 2C and 2D). Population chaos or cycles can be quantitatively detected through the Lyapunov exponent (LE) spectrum (Fig 2B). A positive LE indicates that the population undergoes chaotic dynamics ($g=0.28$, case III; Fig 2B). As the value of $g$ is increased, the population dynamics of $F_1$ and trait dynamics ($\theta_1$) will change from chaos to period-doubling oscillations ($g=0.34$, case IV; Fig 2A, 2C and 2D) and simple cycles ($g=0.48$, case V; Fig 2A, 2C

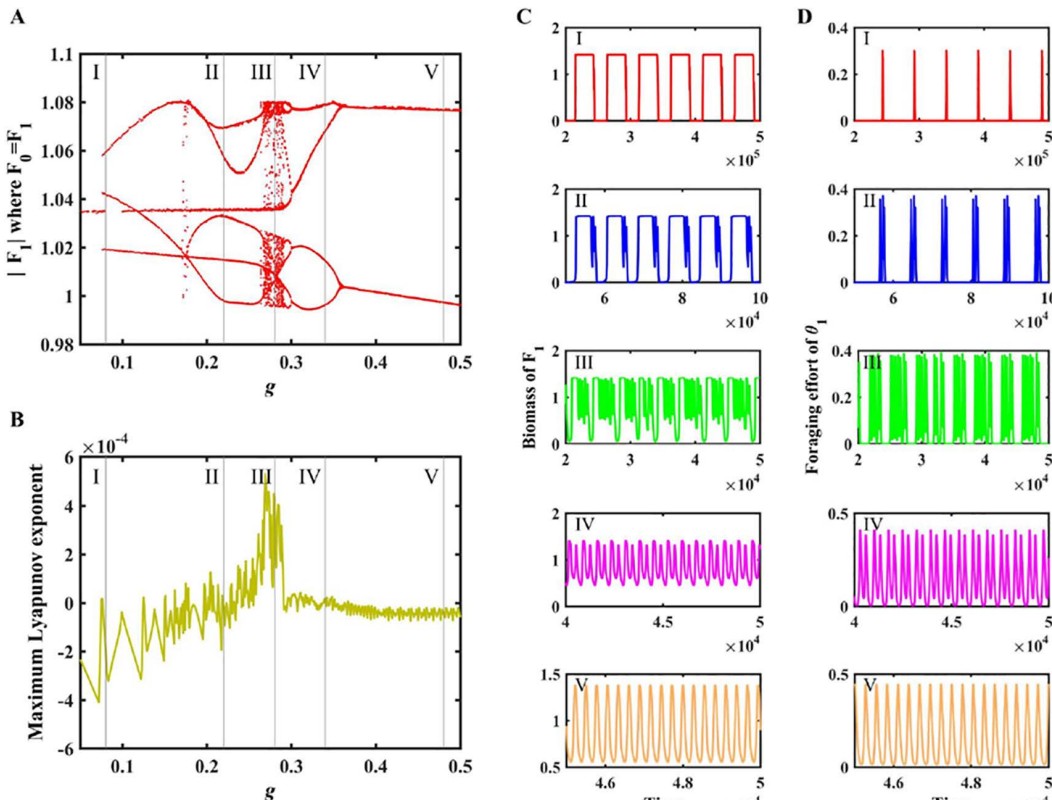

**Fig 2. Intermediate intensity of foraging adaptations generates chaos in the mutualist-exploiter-specialist predator-top predator (i.e., four-species MEST) community.** (A) bifurcation diagram; (B) Lyapunov exponent spectrum; (C) time series of population biomass ($F_1$); (D) time series of foraging effort ($\theta_1$). Model parameters: $r_1=0.35$, $d_1=0.05$, $u_1=0.15$, $\beta_{10}=\beta_{01}=\beta=0.18$, $g \in [0.05, 0.5]$ and other parameter values are presented in Table 1.

and 2D). Overall, neither too high (cases IV and V; Fig 2) nor too low (cases I and II; Fig 2) intensity (*g*) of foraging adaptations lead to chaos, while only intermediate intensity causes the system to generate chaotic dynamics (case III; Fig 2). Furthermore, population dynamics of other species (i.e., $F_1$, *C* and *P*) in the four-species MEST community present similar patterns (Figs A-C in S1 Appendix).

When fixed the intensity (*g* = 0.28; an intermediate adaptation intensity presented in Fig 2) of foraging adaptations, while varying both the consumption rate ($u_1$) and interspecific competition ($\beta_{10} = \beta_{01} = \beta$), we can obtain four food-web structures and their local stability (Fig 3).

First, stable coexistence (i.e., $Re(\lambda_{max})$<0) will not exist ($Re(\lambda_{max})$>0; Fig 3A) when the top predator *P* does not prey on the specialist predator *C* ($\theta_C = 0$); when *P* preys on *C* ($\theta_C \neq 0$), stable coexistence presents at high consumption rate $u_1$ and high interspecific competition $\beta$ ($Re(\lambda_{max})$<0; Fig 3D). Predation of *C* by *P* promotes stable coexistence of species also presented in other food web structures (Fig 3B and 3C). Specifically, when *P* does not prey on the mutualist $F_0$ and *C* ($\theta_C = 0$, $\theta_0 = 0$), no stable coexistence exists in the system ($Re(\lambda_{max})$>0; Fig 3C); when *P* preys on *C* but not $F_0$ ($\theta_C \neq 0$, $\theta_0 = 0$), stable coexistence of species will be achieved over a wide parameter space ($u_1$ and $\beta$): when $\beta$ is low, species coexistence is widespread; while when $\beta$ is an intermediate value, species coexistence is only achieved when $u_1$ is a moderate value ($Re(\lambda_{max})$<0; Fig 3B). Second, by comparing Fig 3B and 3D, we can intuitively find that compared to *P* preys on $F_0$ ($\theta_0 \neq 0$; Fig 3D), P does not prey on $F_0$ will show more stable regions of species coexistence ($\theta_0 = 0$; Fig 3B). Similarly, predation of $F_0$ by *P* is detrimental to community stability and is also presented in other food web structures (Fig 3A and 3C). Specifically, compared to *P* preys on $F_0$ but not *C* ($\theta_0 \neq 0$, $\theta_C = 0$; Fig 3A), P does not prey on $F_0$ and *C* ($\theta_C = 0$, $\theta_0 = 0$) will show oscillations with a lower amplitude ($Re(\lambda_{max})$<0.03; Fig 3C). Finally, to test the effect of parameter selection on the state variables (i.e., $F_0$, $F_1$, C, P, $\theta_0$ and $\theta_1$) in the four-species MEST community, a sensitivity analysis of each parameter was conducted (Table A and Figs A-B in S2 Appendix).

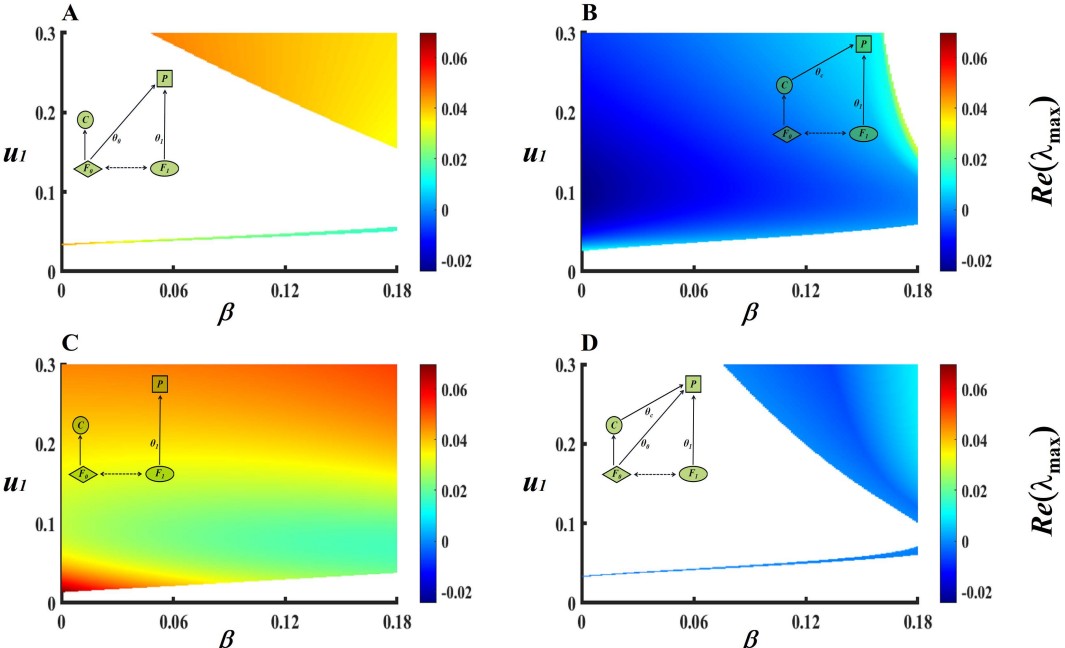

**Fig 3. Network structures and local stability change with the interspecific competition (β) and consumption rate (*u1*) in the four-species MEST community.** Stable coexistence of species is achieved when $Re(\lambda_{max})$<0. In each simulation case, the blue regions have higher stability than the red regions, and the empty white regions denote no solution to network modules. Key parameters of the four-species model: *g* = 0.28, $u_1 \in$[0, 0.3], $\beta_{10} = \beta_{01} = \beta \in$[0, 0.18], $r_1$ = 0.35 and other parameter values are presented in Table 1.

## Foraging adaptation and stability in general MEST communities

In this section, varying adaptation intensity ($g$), interspecific competition ($\beta_{ij} = \beta_{ji} = \beta$) and network size ($N$) could promote system stability (i.e., community persistence at steady state) as a function of the connectance in general MEST communities (Fig 4).

First, when the interspecific competition is low ($\beta = 0.1$; interspecific competition $\beta$ is lower than intraspecific competition (the density-dependence parameter $\alpha_0 = 0.13$)) and the adaptation intensity is fixed ($g = 0.08$), the connectance-stability relationship will go through positive monotonic (i.e., $N = 5$), negative monotonic (i.e., $N = 10, 15, 20$) and peaked (i.e., $N = 25, 30, 35$) patterns as the network size ($N$) increases (Fig 4A). This pattern still exists as the adaptation intensity increasing ($g = 0.28, 0.48$; Fig 4B and 4C). In addition, when the interspecific competition continues to increase ($\beta = 0.12$) and the adaptation intensity is relatively high ($g = 0.28, 0.48$), we could still observe similar patterns of positive monotonic, negative monotonic and finally single-peaked (Fig 4E and Fig 4F). When the interspecific competition is an intermediate value (i.e., $\beta = 0.14 \approx \alpha_0$) and the adaptation intensity is relatively high ($g = 0.28, 0.48$), the connectance-stability relationship mainly shows a peaked pattern (Fig 4H and 4I), while a negative monotonic pattern (Fig 4D and 4G) could be obtained when the adaptation intensity is low ($g = 0.08$) and the interspecific competition approaches the intensity of intraspecific competition $\alpha_0$ (e.g., $\beta = 0.12, 0.14$). Finally, when the intensity of interspecific competition is high (i.e., $\beta = 0.16 > \alpha_0$), as the network size ($N$) increases, a double-peaked pattern of the connectance-stability relationship could be obtained when both the network size and adaptation intensity are relatively high ($g = 0.28, 0.48$ and $N \geq 15$; Fig 4K and 4L), while the connectance-stability relationship does not show a clear trend when the adaptation intensity is small ($g = 0.08$; Fig 4J).

## Comparison between theoretical prediction and empirical data analysis

Given the positive correlation between salt concentration and competition intensity [48], compared to freshwater communities, we deduce that marine communities have higher salt concentrations and are therefore more competitive [8]. On this basis, we compare theoretical prediction with empirical data (i.e., freshwater and marine communities) analysis (Fig 5).

For one thing, model simulations show that the connectance-stability relationship presents a negative monotonic pattern when the interspecific competition is low ($\beta = 0.1$) and the network size is fixed ($N = 15$), while changing the intensity of foraging adaptation ($g$) will not change the decreasing trend (Fig 5A). Our theoretically negative complexity-stability relationship is consistent with empirical result revealed in freshwater communities (Fig 5C and S5 Appendix). For another, when the competition intensity is high ($\beta = 0.16$) and the network size is fixed ($N = 15$), the connectance-stability relationship shows peaked patterns (Fig 5B), i.e., a shift from a single-peaked pattern under the low intensity of foraging adaptation ($g = 0.08$) to a double-peaked pattern under the high intensity of foraging adaptation ($g = 0.48$). Our theoretically peaked patterns may be consistent with data analysis revealed in marine communities (Fig 5D and S5 Appendix).

## Feedback loops due to foraging adaptation regulate community stability

Finally, we propose a potential mechanism of maximal positive or/and negative feedback loops [49] for the foraging adaptation affecting the stability of the MEST community (Fig 6).

In the four-species MEST community (i.e., network structure in Fig 3D), when adaptive foraging ($AF$) behaviour is not considered, the network has two maximal feedback loops (Fig 6A): positive feedback loop $\left( F_0 \overset{+}{\to} C \overset{+}{\to} P \overset{-}{\to} F_1 \overset{-}{\to} F_0 \right)$ and negative feedback loop $\left( F_0 \overset{-}{\to} F_1 \overset{+}{\to} P \overset{-}{\to} C \overset{+}{\to} F_0 \right)$. When the $AF$ is introduced into the four-species system, the new system has three types of maximum feedback loops based on trait-mediated indirect effects (i.e., foraging preference trait of the top predator modulates the system stability by altering the dominant positive or/and negative feedback loops): i) negative feedbacks (Fig 6B), where the foraging strategy is to increase predation on the specialist predator

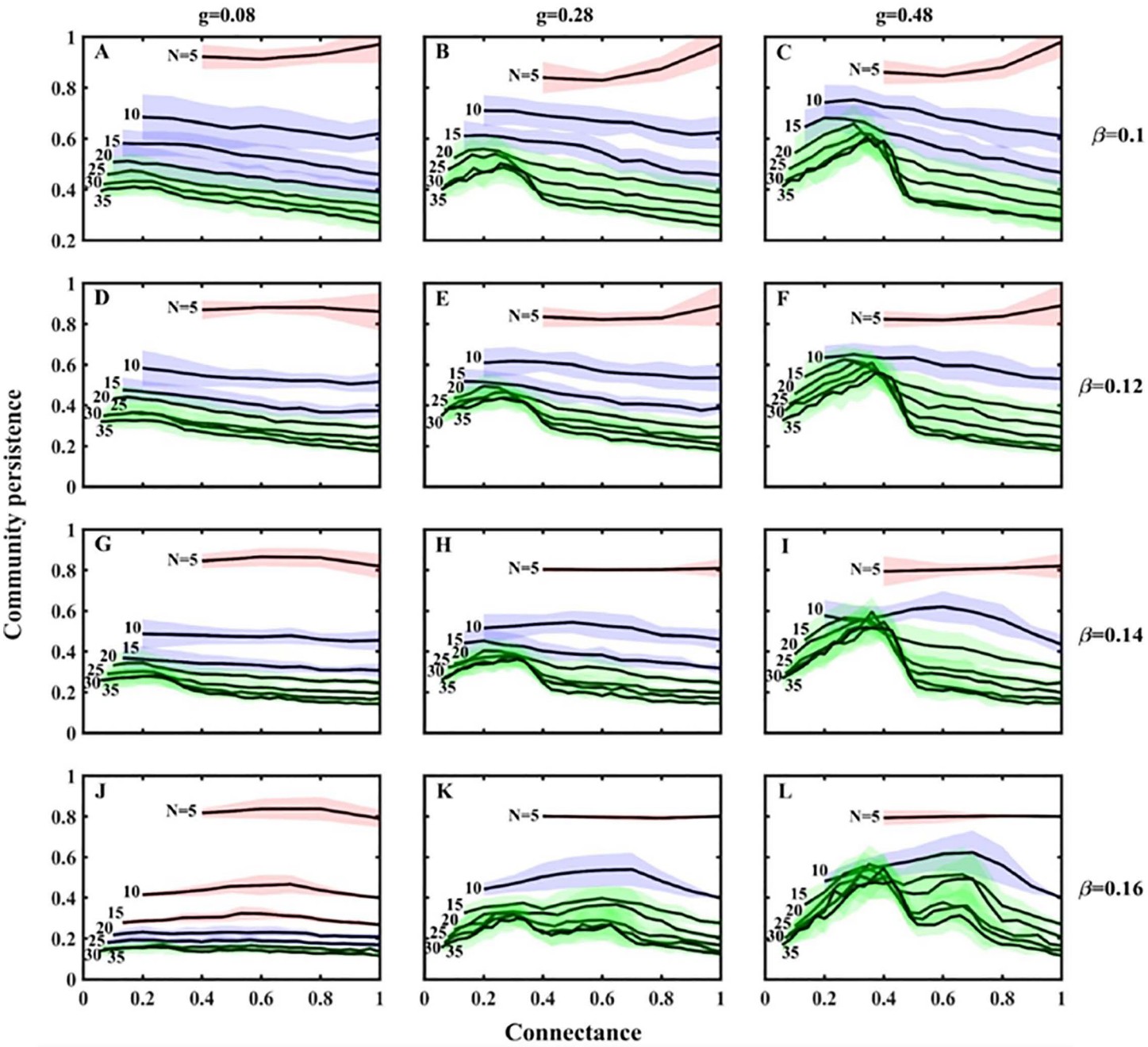

**Fig 4. The variable relationship between initial connectance and community stability (community persistence at steady state; mean ± standard deviation based on 20×20 replicates) in general MEST communities.** The different coloured regions represent different patterns of connectance-stability relationships. Key parameters of the full model: $r_i \sim U[0.3, 0.4]$, $d_i = 0.05$, $u_i = 0.15$, $\beta_{i0} = \beta_{ji} = \beta$, and other parameter values are presented in Table 1.

$C$ ($AF$ inhibits $C$) and reduce predation on the exploiter $F_1$ ($AF$ promotes $F_1$), and we can obtain two negative feedback loops $\left(P \xrightarrow{+} AF \xrightarrow{-} C \xrightarrow{-} F_0 \xrightarrow{-} F_1 \xrightarrow{+} P \;\&\; P \xrightarrow{+} AF \xrightarrow{+} F_1 \xrightarrow{-} F_0 \xrightarrow{+} C \xrightarrow{+} P\right)$; ii) positive feedbacks (Fig 6C), where the foraging strategy is to reduce predation on $C$ ($AF$ facilitates $C$) and increase predation on $F_1$ ($AF$ inhibits $F_1$), and we can

obtain two positive feedback loops $\left( P \xrightarrow{+} AF \xrightarrow{+} C \xrightarrow{-} F_0 \xrightarrow{-} F_1 \xrightarrow{+} P\ \&\ P \xrightarrow{+} AF \xrightarrow{-} F_1 \xrightarrow{-} F_0 \xrightarrow{+} C \xrightarrow{+} P \right)$; iii) positive and negative feedbacks (Fig 6D), where the foraging strategy is to reduce (or increase) predation on both $C$ and $F_1$, and we can obtain two groups of both positive and negative feedback loops: the increased pre-dation case $\left( P \xrightarrow{+} AF \xrightarrow{+} C \xrightarrow{-} F_0 \xrightarrow{-} F_1 \xrightarrow{+} P\ \&\ P \xrightarrow{+} AF \xrightarrow{+} F_1 \xrightarrow{-} F_0 \xrightarrow{+} C \xrightarrow{+} P \right)$ and the decreased predation case $\left( P \xrightarrow{+} AF \xrightarrow{-} F_1 \xrightarrow{-} F_0 \xrightarrow{+} C \xrightarrow{+} P\ \&\ P \xrightarrow{+} AF \xrightarrow{-} C \xrightarrow{-} F_0 \xrightarrow{-} F_1 \xrightarrow{+} P \right)$. For general MEST communities, the maximum feedback loop of the network without the $AF$ also includes both positive and negative feedbacks; while when the $AF$ is introduced into the community, the maximum feedback loop for the new system is similar to the four-species case presented in Fig 6.

## Discussion

In this study, we develop an adaptive network model to evaluate the connectance-stability relationship in mutualist-exploiter-predator communities. Our results showed that adaptive foraging ($AF$) contributes to the stability of mutualism and intermediate intensity of foraging adaptations can lead to chaotic dynamics. Moreover, the connectance-stability

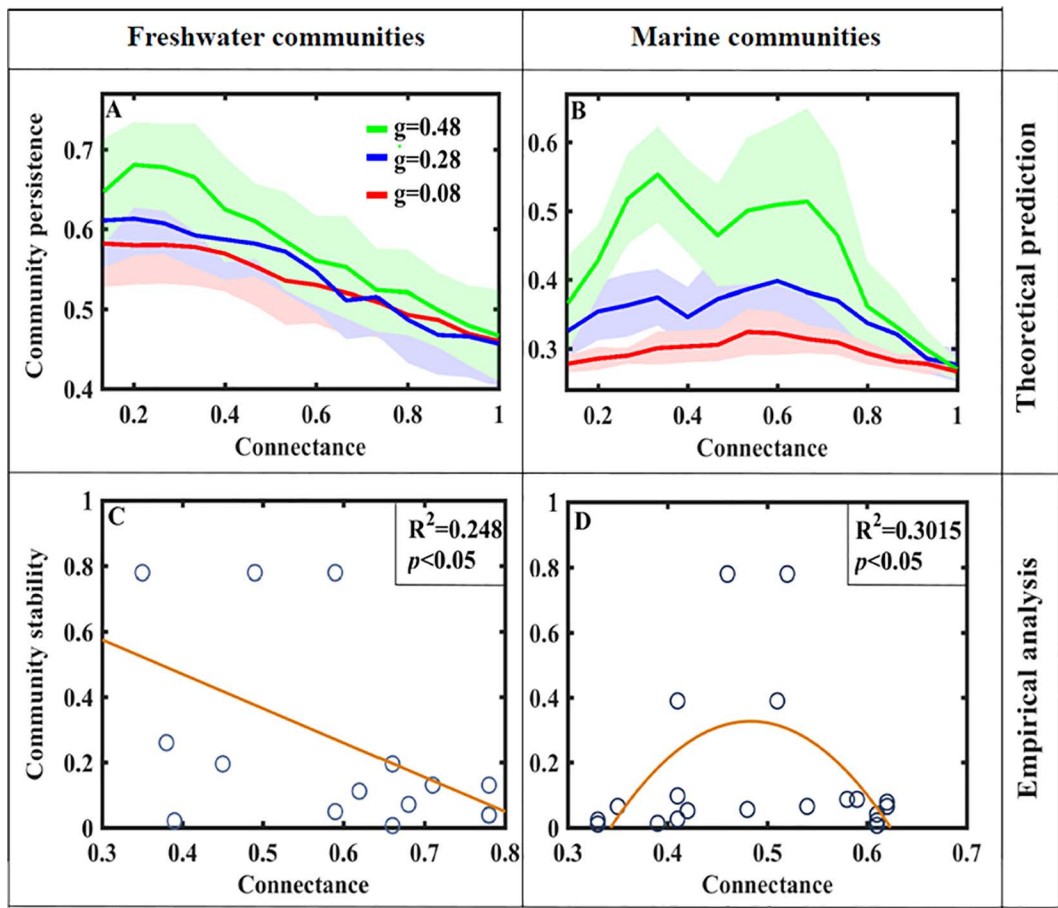

**Fig 5. Comparison between theoretical prediction and empirical data analysis.** (A, B) theoretical prediction presented in Fig 4; (C, D) empirical data analysis presented in the previous study [8], while the community stability is measured as $0.02/Re(\lambda_{max})$ for (C) freshwater communities and (D) marine communities. The fitted curve in (C) Stability = 0.89125 − 1.05279 × Connectance; in (D) Stability = 16.04436 × Connectance − 16.60223 × Connectance² − 3.54895.

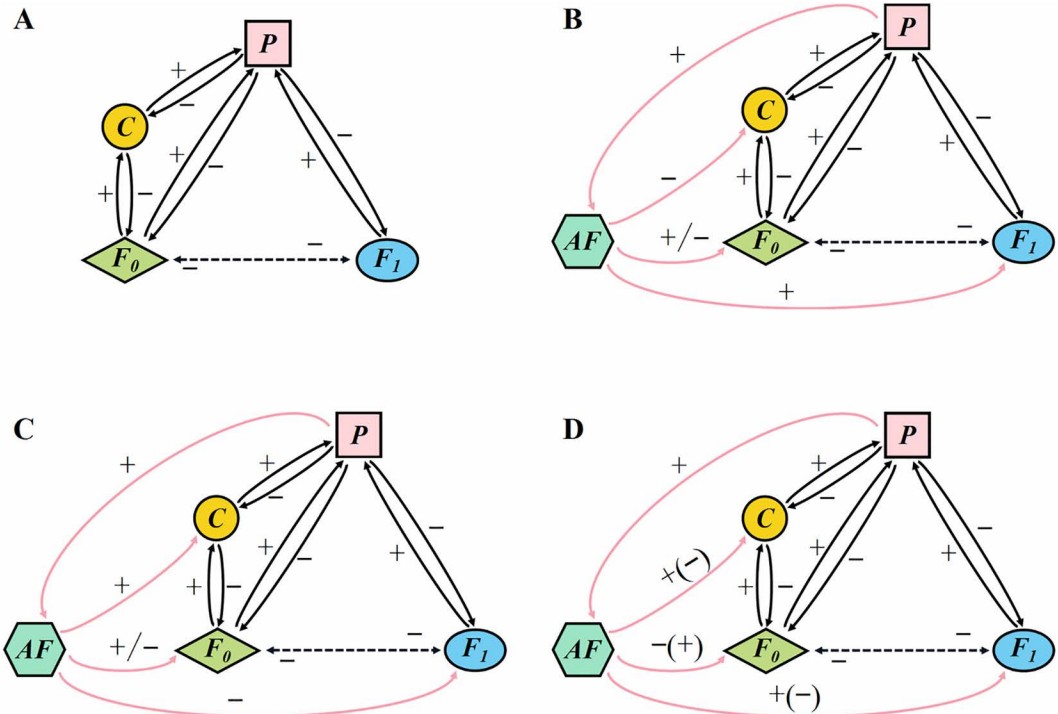

**Fig 6. Foraging adaptation regulates the stability of the MEST community by changing positive or/and negative feedback loops.** Sign '+' indicates promotion and '-' indicates inhibition. AF-adaptive foraging. (B, C) '+/-' includes one case: promotion or inhibition; (D) '+(-)' and '-(+)' include two cases: promotion and inhibition, respectively.

relationship may show different patterns (i.e., positive monotonic, negative monotonic, peaked and double-peaked patterns), while the double-peaked pattern could be obtained when both the adaptation intensity and interspecific competition are large. In addition, our theoretical predictions may be consistent with empirical results revealed in both freshwater and marine ecosystems. Finally, we inferred that the foraging adaptation alters positive or/and negative feedback loops (trait-mediated indirect effects) to affect community stability.

## Intermediate intensity of foraging adaptation generates chaos

Our theoretical results found that increasing the intensity ($g$) of foraging adaptation, the system would switch between periodic and chaotic states (Fig 2); moreover, such chaotic dynamics may also emerge in a relatively complex community (Fig A in S4 Appendix). Our work supports the earlier theoretical view of foraging at the edge of chaos [50,51], which operates in a critical dynamical mechanism between order and chaos [52]. Indeed, ecosystems, as complex adaptive systems [53], may stay at the edge of chaos [54]. In addition, our theoretical results suggested that only the intermediate intensity of foraging adaptation can lead to chaos (case III; Fig 2), whereas too low (cases I and II; Fig 2) or too high (cases IV and V; Fig 2) cases only generate regular/periodic dynamics. Earlier theoretical and empirical work showed that chaos may provide a means for flexibility [55,56]. In our system, an intermediate adaptation intensity of the top predator can produce the flexibility, which may facilitate the adaptation of the food web to environmental disturbance. This finding coincides with earlier studies of food webs [57], fig-wasp mutualism [47] and antigenically diverse pathogens [58], which reveal that the moderate levels of stress may lead to chaos or complexity maximization.

## Adaptive foraging contributes to the stability of mutualism

We found that predation by the top predator on the specialist predator will promote stable coexistence (Fig 3B and 3C), while predation by the top predator on the mutualist will greatly reduce the area of stable coexistence (Fig 3B and 3D). In other words, the protection of mutualist by top predator (non-predation on mutualist or predation on the specialist predator of the mutualist) will contribute to community stability. This finding supports one early research of mutualistic networks [59], which showed that incorporation of adaptive foraging (AF) into the dynamics of mutualistic systems increased the community persistence. Moreover, our work also supports earlier empirical work [37], which revealed that predatory ants (P) contribute to the maintenance of mutualism between the fig tree (i.e., the basal resource) and pollinating fig wasp ($F_0$). However, our results reverse recent theoretical findings [29]. The latter also studied the four-species MEST community without the AF, and found that the top predator (P) would always become extinct when omnivorous link does not exist (i.e., $\theta_0 = 0$; please also see the cited work presented in Fig 2i and the diagonal of region $C_{5b}$ in Fig 5). Similar to evolutionary rescue theory [60,61], our adaptive network model reveals that the AF of the top predator may rescue itself from extinction by comparing the no-foraging adaptive scenario [29]. Finally, our model results also reverse theoretical predictions presented in the resource-mutualist-exploiter-predator community, the latter found that stable coexistence (i.e., $Re(\lambda_{max}) < 0$) may not emerge when the top predator P does not prey on the mutualist F0 (Fig A in S3 Appendix). An important reason for the inconsistency is that the new theoretical framework neglected life history of the basal resource, while the life cycle of fig trees (R) is much longer than that of fig wasps (i.e., $F_0$, $F_1$, C) and the top predator (i.e., P). Therefore, compared with directly adding the basal resource to the model (S3 Appendix), it is more reasonable to ignore the resources and regard the mutualist as one potential resource (Fig 3).

## Network size and interspecific competition regulate community stability

In our recent work [29], we found that the increased competition can promote species coexistence in the omnivorous food web (the four-species MEST community); similarly, in this study, we also revealed that positive correlation between interspecific competition and community stability under the intermediate to large network sizes (Fig 4): when we fix the network size and connectance (N = 10 and C = 0.7; blue regions in Fig 4C, 4F, 4I and 4L), increasing the competition intensity (β) will cause stability to first decrease and then increase; when the network size is large (N = 20 and C = 0.7; green regions), the increase of the competition intensity (β) will cause a steady increase in community stability; moreover, when the adaptation intensity is low (g = 0.08; Fig 4A, Fig 4D, 4G and 4J) or an intermediate value (g = 0.28; Fig 4B, 4E, 4H and 4K), the competition-stability relationship shows similar patterns under the fixed network connectance. Our theoretical results support earlier theoretical and empirical work, which show the competition promote species coexistence in a theoretical framework [62] and attributes to stability of culturable microbial species [63].

## Reconcile the complexity-stability debate in empirical ecosystems

Our theoretical predictions may match empirical results to a certain extent, showing a negative monotonic complexity-stability relationship (Fig 5A and 5C and S5 Appendix) and the peaked pattern (Fig 5B, 5D; S5 Appendix); similar to early multiple complexity-stability relationships [22,64,65], our work revealed that network complexity (i.e., connectance) may be not linearly correlated with stability in empirical ecosystems [8]. Moreover, our theoretical results found that the network size may reverse the connectance-stability relationship (Fig 4). This finding is consistent with earlier studies on the effects of the omnivory, diversity, and interaction strength on system stability [15–17]. Both of them found that the increase in system dimension can change the cases from stable to unstable states. Finally, our adaptive network framework that couples the mutualistic-antagonistic interaction may provide some suggestions for empirical work. For instance, experiments should be designed to incorporate the adaptive behaviour of studied individuals into the system (e.g., a good example is the adaptive learning of foraging skills in fish [66]), after all, ecosystems are often considered as complex adaptive systems [53].

### Foraging adaptation and feedback loops regulate community stability

In this study, we treated the foraging preference of the top predator as a behaviour trait and coupled it to population density/biomass (including density-mediated effects and trait-mediated indirect effects [67,68], while the latter can reinforce or oppose the former), and the theoretical results revealed that there may be multiple types of connectance-stability relationships (Fig 4).

One potential reason for these patterns may be that adaptive foraging (AF) of the top predator alters the positive or/and negative feedbacks of the maximum feedback loop (Fig 6), which is often used to characterize system stability [49,69], and thus regulates community stability. In biological systems, positive feedback loops can move the system away from equilibrium (unstable state), while negative feedback loops can reduce the effects of perturbations and return the system to a stable state [70]. In our study, when the top predator ($P$) increases predation on the specialist predator ($C$) and reduces predation on the exploiter ($F_1$) (Fig 6B), the maximum feedback loop is negative feedback, which explains the positive monotonic pattern (Fig 4), and this positive pattern is consistent with early work of food webs [22]; moreover, the maximum feedback loop is positive feedback if $P$ reduces predation on $C$ and increases predation on $F_1$ (Fig 6C), which may explain the negative monotonic pattern (Fig 4), and this negative case coincides with earlier theoretical work [1]. Finally, the other patterns (i.e., peaked and double-peaked patterns; Fig 4) may be caused by alternating dominant effects between positive and negative feedback loops (Fig 6D), and this peaked pattern is in line with previous theoretical study [65]. Specifically, the double-peaked pattern has one large peak and one small peak, and it may be a special case (i.e., damped oscillation) of the peaked pattern. Overall, the top predator changes the positive or/and negative feedback loops of the system by flexibly adjusting foraging strategies (trait-mediated indirect effects), thereby changing community stability [71,72].

## Future direction

Future work will expand on the following two aspects: first, multilayer network structures need to be introduced into the adaptive network framework [27,28,73]. In this study, we focused on the effects of foraging adaptation by a top predator on community stability (i.e., our simulations were drawn from a limited set of examples and exhibited a somewhat restricted structure), whereas a more general scenario may be foraging selection by multiple predators on multiple prey [21,74]. Second, defensive traits of prey should be incorporated into the adaptive network framework [20,75]. For instance, plant secondary metabolites can affect the behaviour, feeding and digestion of pests to achieve resistance to pests [76]. However, in our work, only adaptive foraging (AF) of the top predator was considered and the adaptive behaviour of the prey is ignored. For a more realistic modeling framework, it is necessary to consider the complex adaptive behaviour of the individuals studied in the model [21,77,78]. Overall, coupling the AF of predators, prey defence traits and multilayer network structures to explore the structure-stability relationship in real ecosystems is urgently needed.

## Supporting information

**S1 Appendix. Population chaos in the four-species MEST community.**
(DOCX)

**S2 Appendix. Sensitivity analyses of model parameters in the four-species MEST community.**
(DOCX)

**S3 Appendix. Structural sensitivity analyses of mutualist-exploiter-predator communities with the basal resource.**
(DOCX)

**S4 Appendix. Population chaos and flexible network structures in a complex community.**
(DOCX)

**S5 Appendix. Model fitting based on empirical data.**
(DOCX)

**S6 Appendix. Simulation codes for all figures.**
(ZIP)

## Acknowledgments

We thank Ellen van Velzen and Peter Abrams for helpful comments on an earlier version of this manuscript.

## Author contributions

**Conceptualization:** Lin Wang, Jin-Bao Liao, Rui-Wu Wang.

**Data curation:** Xiao-Wei Zhang.

**Funding acquisition:** Lin Wang, Jin-Bao Liao, Rui-Wu Wang.

**Methodology:** Lin Wang, Ting Wang, Xiao-Fen Lin, Jia Li.

**Software:** Ting Wang, Xiao-Fen Lin, Jia Li.

**Supervision:** Jin-Bao Liao, Rui-Wu Wang.

**Visualization:** Ting Wang, Xiao-Wei Zhang, Xiao-Fen Lin, Jia Li.

**Writing – original draft:** Lin Wang.

**Writing – review & editing:** Lin Wang, Ting Wang, Xiao-Wei Zhang, Xiao-Fen Lin, Jia Li, Jin-Bao Liao, Rui-Wu Wang.

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
