## [Decision Letter · Decision Letter 0]

PCOMPBIOL-D-24-01756

Evolution of foraging behaviour induces variable complexity-stability relationships in a multiplex community with multiple interaction types

PLOS Computational Biology

Dear Dr. Wang,

Thank you for submitting your manuscript to PLOS Computational Biology. After careful consideration, we feel that it has merit but does not fully meet PLOS Computational Biology's publication criteria as it currently stands. Therefore, we invite you to submit a revised version of the manuscript that addresses the points raised during the review process.

Please submit your revised manuscript within 60 days Feb 21 2025 11:59PM. If you will need more time than this to complete your revisions, please reply to this message or contact the journal office at ploscompbiol@plos.org. Please include the following items when submitting your revised manuscript:

We look forward to receiving your revised manuscript.

Kind regards,

Samraat Pawar

Academic Editor

PLOS Computational Biology

Tobias Bollenbach

Section Editor

PLOS Computational Biology

**Additional Editor Comments:**

Thank you for your submission. The manuscript has received mixed feedback from three reviewers. While all three recognize the novelty and potential impact of addressing complexity-stability relationships in ecological networks with adaptive foraging, they highlight critical issues that must be addressed. Reviewer 1 and Reviewer 3 recommend major revisions, focusing on improving model justification, clarity of results, and empirical validation. However, Reviewer 2 raises stronger concerns, including the lack of clear research questions, methodological ambiguity, and ecological relevance, and recommends rejection.

I think that the manuscript could make a significant contribution to the field if these issues are **thoroughly** resolved.

Therefore, I am recommending a major revision supported by point-by-point responses to the reviewers' comments.

In particular,

* Reviewer 1 questions the generalizability of the findings due to the simplistic model structure and asks for clarification on its applicability to larger ecosystems, requests stronger justification for parameter values and suggests robustness analyses (also sharedby other reviewers), and criticizes the inconsistency in how stability is measured, noting the shift from eigenvalue-based analysis to community persistence in larger networks. They also recommend improving the clarity of terms, figure labeling, and technical definitions.

* Reviewer 2 criticises the lack of clear objectives and hypotheses, unclear / unjustified methodological details, and most importantly, highlights issues with the model: exclusion of basal resources, decoupling of competition dynamics, and parameter sensitivity (again). They also note that empirical validation with freshwater and marine systems is poorly integrated and inadequately explained (again, shared by Rev 3).

Reviewer 3 criticises the lack of clarity in linking different model findings and suggests additional simulations to control of confounding factors (so the same theme of parameter and structural sensitivity and robustness analysis again), and asks for more rigorous statistical approaches in the empirical validation and better explanation of the modification of connectance. They also highlight (and again a common theme) difficulties in following the choice of parameter variation and understanding which modeling elements drive specific results, and suggest enhancing figure clarity and consistency to aid interpretation.

Overall, therefore I suggest that your revision addresses the following in particular:

* Clearly state the research question(s) and hypotheses in the introduction (Reviewer 2).

* Clarify/justify key assumptions and such as parameter values, and provide robustness or sensitivity analyses (Reviewers 1 & 2).

* Address model limitations (structural sensitivity), by incorporating basal resources, or justify their exclusion analytically.

* Explain the treatment of competition dynamics and analyse the coupling of intra- and interspecific competition (Reviewer 2).

* Provide detailed descriptions of empirical model fitting and comparison, and clearly explain the relationship between model predictions and observed data (Reviewers 2 & 3).

* Reconcile the use of different stability metrics and justify their selection for specific network sizes (Reviewer 1).

* Enhance figure consistency (e.g., color schemes) and labeling for clarity, and reorganize the manuscript to place the methods section earlier (Reviewers 1 & 3).

* Strengthen the connection between model dynamics (e.g., feedback loops) and observed stability trends through additional analyses or simulations (Reviewer 3).

If you do choose to undertake the revision, the final decision will depend on the point-by-point responses, revisions, and potentially, further review (ideally by the same reviewers).

**Journal Requirements:**

1) We noticed that you used the phrase 'not shown' in the manuscript. We do not allow these references, as the PLOS data access policy requires that all data be either published with the manuscript or made available in a publicly accessible database. Please amend the supplementary material to include the referenced data or remove the references.

2) We have noticed that you have uploaded Supporting Information files, but you have not included a list of legends. Please add a full list of legends for your Supporting Information files after the references list.

3) Please ensure that the funders and grant numbers match between the Financial Disclosure field and the Funding Information tab in your submission form. Note that the funders must be provided in the same order in both places as well." Currently, the order of these grants "32201264" ,32271548" and "U2102221" is different in both places.

Please indicate by return email the full and correct funding information for your study and confirm the order in which funding contributions should appear. Please be sure to indicate whether the funders played any role in the study design, data collection and analysis, decision to publish, or preparation of the manuscript.

**Reviewers' comments:**

Reviewer's Responses to Questions

Reviewer #1: This manuscript explores how adaptive foraging behavior in complex ecological networks influences community stability in systems with both mutualistic and antagonistic interactions. Using an adaptive network model, the authors demonstrate that changes in foraging preferences can result in diverse stability patterns, including single and double peaks, potentially explaining the varying stability trends observed in real-world ecosystems, such as freshwater and marine environments. The proposed adaptive framework holds promise for capturing both observed patterns and differences in the complexity-stability relationships across these systems.

This manuscript offers a novel theoretical framework with valuable insights into the complexity-stability debate, which has the potential to make a meaningful contribution to the field. However, there are several areas where significant revisions are needed. In particular, I believe the authors should provide stronger justification for their model assumptions, clarify the treatment of stability metrics, and strengthen the connections to empirical systems. I outline three major concerns along with a list of minor issues, which I hope the authors will find constructive in improving the manuscript.

Major Issues

1. Simplistic Community Structure:

The manuscript focuses on a simple omnivorous food web incorporating mutualistic and antagonistic relationships with only four species across three trophic levels: top predator, specialist predator, mutualist, and exploiter. While such a setup is useful for tractability and analysis, it raises questions about the generalizability of the findings to real-world ecosystems. Specifically, what would happen if there were more than one specialist predator or more layers of interactions? The authors need to justify how their framework applies to larger, more realistic ecological systems.

Additionally, it is unclear how this model relates to the two empirical systems—freshwater and marine—mentioned in the paper. The primary difference between the two empirical communities is the intensity of interspecific competition, which is deduced in this manuscript. This difference leads to variations in adaptive responses and ultimately drives the contrasting stability outcomes between freshwater and marine systems. However, this difference requires further justification.

2. Parameter Values:

The parameter values in Table 1 lack adequate justification. For instance:

- Why is the growth rate of F0 set at 0.5? Is this value based on empirical systems? If so, the authors should specify which system. If not, a robustness analysis testing other growth rate values would be necessary.

- Why are the growth rates of Fi always smaller than those of F0? This assumption should be explained, particularly in the context of ecological realism.

- The conversion efficiency is set at 1, implying perfect conversion. This is an idealized assumption that may not hold in real-world systems. The authors should clarify why this assumption was made and discuss its potential impact if it does not hold.

- The density-dependent coefficients (alpha0, alphaC, alphaP) are set at 0.1, 0.12, and 0.13, respectively, but there is no clear reasoning for these specific values. The manuscript would benefit from an explanation of why these coefficients vary in this way and how sensitive the model outcomes are to changes in these parameters.

- The intensity of foraging adaptation and competition parameters also require clearer biological or ecological justification. Without this, it is difficult to assess the model’s applicability to real systems.

3. Inconsistency in Stability Metrics:

There is a fundamental inconsistency in how stability is defined and measured in the manuscript. For small networks, the authors use local stability analysis based on eigenvalues, which measures how quickly a system returns to equilibrium after perturbations. This is a classic and well-established approach.

For larger networks, however, the authors shift to using community persistence (the fraction of remaining species after long simulations) as a proxy for stability. This measure is more aligned with species diversity rather than stability, as it does not capture dynamic responses to perturbations. Given the long-standing debate on the diversity-stability relationship, extreme care must be taken when using diversity-related measures as a proxy for stability.

The manuscript should clearly differentiate between these two metrics and their implications. Unless additional analyses are performed to measure stability in larger communities (e.g., analyzing eigenvalues or resilience metrics), the claims about stability in large networks are not fully convincing.

Minor Issues

1. Some terms, such as “positive-negative feedback,” could be simplified or better explained.

2. Figures, particularly Figures 2 and 3, would benefit from consistent labeling and coloring for stability regions to make the results easier to interpret.

3. Including brief definitions for technical terms like "trait-mediated indirect effects" and "Holling type I functional response" would help readers less familiar with ecological modeling.

4. In line 456, the equation mentions "consumption of C" instead of F0. This seems inconsistent with the context and should be clarified.

5. While the GitHub code link is provided, a README file with instructions on how to run the code would make it more accessible to researchers who want to replicate or build upon the study.

Reviewer #2: In the current study entitled “Evolution of foraging behaviour induces variable complexity-stability relationships in a multiplex community with multiple interaction types”, the authors, Wang et al. extend from their previous work to create a multi-species (i.e., beyond four species) version the MEST model, which includes a top predator that feeds on all others, a specialist predator that feeds on the mutualist, as well as a mutualist and one or more exploiters that feed on a shared resource. The goal of the study is to explore the influence of foraging preference evolution (in terms of the top predator’s foraging effort allocation among prey species) on the system’s stability (in terms of local stability, evaluated by eigen-value analysis of the Jacobian, and community persistence, in terms of the number of species that remain after a long run) when the system has both antagonistic and mutualistic interactions. This research direction is inheriting, and combining the topics, explored by M. Kondoh (ref 13 and 29 in the manuscript).

While the study indeed targets an existing gap in the field of complexity-stability debate (i.e., the effect of adaptive foraging with mix-type interactions), and the reported behaviours of the model (especially the uni- and bi-modality in the complexity-stability relationships) are somewhat interesting, unfortunately, I find the manuscript in its current form rather unsatisfying, and this is due to a combination of the arbitrariness of the modelling part, the ambiguity of the overall methodological description, and thus, the unconvincing discoveries and relevant ecological interpretation. Below I elucidate my major concerns, and I hope this could help the authors to improve the study.

1. There is no clear research question nor hypotheses stated, and therefore the readership would not know what to expect in this manuscript even after reading the introduction---is the current study going to manipulate network structure, the composition of antagonistic vs. mutualistic elements, interaction strengths, or other key parameters? What is going to be “tested” after all? The journal PLOS Computational Biology emphasises reliability and significance of biological discovery through computation, so a sound framing of the research question, and biologically relevant hypotheses, are definitely needed.

2. Following the previous point, some of such information are only given in the methods section, however without reasoning or justification, and relevant details are only very loosely described. So, even after reading the methods, the readership would not know why the authors decided to test such aspects of the model, and why manipulated in such ways. Some examples are:

2-1. There is no notice nor reasoning for that the authors are testing 4-sp and multi-sp models separately. Are these two parts answering different questions? What is the goal for extending 4-sp to multi-sp? None of these are explained.

2-2. When moving to the multi-sp model, why it is only the number of exploiters being increased, but not the other roles? Is it for mimicking a realistic system (e.g., the wasp system in Fig. 1A)? But if so, is the goal of the study to specifically address the dynamics in that system, or to tackle a very general mechanism in ecological dynamic systems? From the introduction it sounds like the latter; yet from the methods, and the fact that the authors associate the findings to freshwater and marine food webs, not so.

2-3. And, speaking of the empirical validation with freshwater and marine food webs, this part is not mentioned or explained anywhere at all. It just appears out of the blue in the results, without any context. What models are those fitted against the data? What are the alternative models and are there any model selection process? Wouldn’t a unimodal model also (maybe better) fit to the data of Fig. 6C? Having this bit in the manuscript does not help make the model more convincing, instead incur confusions.

2-4. L495: the description here itself does not give any detail regarding how the authors derive the equilibrium of a dynamic system with adaptively changing interaction strengths. And thus, it is unclear at what states the eigenvalues are evaluated, and how such states are derived/approached.

2-5. L498 onwards: the authors state that “it is more difficult…” so that they use an alternative way (remaining sp after a long run) of quantifying stability in the multi-sp model other than the eigenvalue. I do not quite get this reasoning. What are the boundary equilibrium points, and why is it difficult to list all possible network structures? How large is the network mentioned here after all---it is not mentioned; I can only tell from Fig. 3 that the authors simulated network sizes from 5 to 35 (which, again, is a setting not mentioned nor justified elsewhere), yet 35 does not seem like a very big network. I do not get why the approach doable to a 4-sp system (which, as pointed out in the previous point, is not explained) cannot be applied to a 35-sp system, especially when all the species added to the system are of the same role (exploiters) so the complexity is actually constrained.

2-6. L513: Why highlighting specifically a N = 9 case? Is this size specifically representative or ecologically relevant? Why not other sizes?

2-7. Why the main focus is on beta, g, and N, but not other aspects/parameters? What’s the rationale of this decision? And how the investigation is carried out---e.g., in terms of the range of values that the authors explore?

2-8. Table 1: How are these values determined, and what are the units of them? Without any justification, I can hardly believe that these values are within a biologically meaningful or realistic range. Also, seemingly there is no sensitivity analyses on the simulations at all? If all the conclusions are drawn from only one set of parameters, wouldn’t all the discovered patterns are in fact outcomes that are sensitive to the parameter chosen, rather than that of general ecological meaning?

3. My biggest concern lies on the model itself.

3-1. Why not including the basal resource, as indicated in Fig. 1A, as a state variable into the model? Without this entity, the model ignores several important feedbacks in the system. For instance, the mutualist in the model in effect is not a “mutualist” at all in the current system---there is no other entity that is “mutually benefitting” (+/+) with it. As illustrated in Fig. 5A, the interactions between the mutualist and the exploiters are effectively competition (-/-) and are incorporated into the model’s interaction matrix just as all other interactions; however, these are actually indirect interaction and should be mediated through the abundance of a basal resource, but such resource-dependency (which would be a constraining coupling between direct and indirect interaction strengths) is excluded from the model.

3-2. As stated above, I am therefore very doubtful regarding all the patterns reported. If, as the authors appear to claim, that those patterns could be of ecological generality and help resolve different complexity-stability relationships in the literature, then I would expect to see the same patterns emerge from some alternative models where a basal resource is incorporated as a state variable, no matter as a species or a chemostat flux. It would be very helpful if the authors could proof this by conducting additional simulations with new models.

3-3. Additionally, in the current model, interspecific and intraspecific competition are decoupled, too. For the exploiters, their intraspecific competition is modelled via a mutualist-dependent carrying capacity in the logistic term (though independently for each exploiter, determined by a shared parameter q), and interspecific competition as separated terms. However, these two should be coupled and modulated by the abundance of the basal resource, too. The current model without a basal resource therefore ignores the feedback created by a change in exploiters’ abundances to their own growth (via decreasing the resource). Such decoupling of resource-driven intraspecific and interspecific competition could create unrealistic coexistence purely due to stronger former than the latter and determined by the authors’ parameter chosen, while these two should coupled in scales via their mechanistic resource dependence. I thus have no much faith in the coexistence patterns revealed.

3-4. A relatively minor point: please clarify the direction of effect in parameter beta_ij, is it the effect of j on i, or the reverse? Since the whole simulation and analysis structure is mirroring the work by M. Kondoh, though with Kondoh’s annotation the direction of the effect seems to be the opposite of the current study. I am therefore concern if the interaction matrix is corrected coded (could not check the codes though, apologies---am not a matlab user).

Reviewer #3: SUMMARY AND GENERAL IMPRESSIONS

The present manuscript analyzes a model of a mutualist, exploiters and two types of predators (motivated by an empirical wasp system) using numerical simulations. In addition to the food web structure, the top predator adaptively forages on all other species. The manuscript first determines the stability diagram for the four species food web structure and then adds many exploiter types to the model that vary in growth rates and network connections. In this context, it is shown that the stability of the entire system can have several dependencies on the connectance of the food web, including non-monotonic relationships. Two related observations are presented. First, there are chaotic dynamics in the model, and second, a number of new feedback loops emerge when the top predator adaptively forages. Last, the authors argue that empirical data from freshwater and marine systems display some of the behavior they found through simulations.

I found the stability-connectance results counterintuitive and interesting. Similarly, the chaotic dynamics have a beautiful mathematical structure. In my view, the connection between these two sets of results needs to be made stronger. Because there are many processes in the model, it is difficult to know what is causing the new stability-connectance results. It would be useful to have some clearer “control” simulations in which we can build the counterintuitive results from each of the modeling ingredients. In the Technical Comments section below, I list a number of more specific comments related to this concern. The results in the manuscript could be connected to one another more clearly and the analysis of the empirical data is lacking. In addition, the readability of the manuscript could be improved. It can be difficult to follow exactly which parameters are being varied on a first reading. Similarly, it is difficult for the reader to build an intuition about what elements of the model are producing the results. The Readability Comments section has more specific comments describing these issues.

TECHNICAL COMMENTS

As I understand it, the paper is arguing that adaptive foraging is causing new stability-connectance relationships via non-equilibrium dynamics, but the connections between these different sets of results is tenuous. First, it would be useful to have a panel in Fig. 3 without adaptive foraging to convince the reader that adaptive foraging is a crucial ingredient for the new behaviors. Is it possible to derive the behavior of a model without adaptive foraging analytically using the standard complexity-stability approach? Results for a simple model without adaptive foraging will provide a baseline for the reader to understand which behaviors come from which processes.

Second, if the authors intend to argue that the non-equilibrium behaviors they observe cause the surprising stability-connectance behaviors, there needs to be a stronger quantitative connection between them. For example, do the peaks in the non-monotonic behavior in Fig. 3 correlate with the frequency of simulations that are not at equilibrium or exhibit chaos? It might be instructive to analyze some simpler scenarios. For example, if every exploiter has the same growth rate and a simple network structure is chosen, perhaps it is possible to draw some analytical conclusions.

From Equation (5) in the Methods, it appears that g cannot change the equilibrium solutions of the model, though it could affect stability. If this is true, then it is additional evidence that adaptive foraging creates non-equilibrium solutions.

Relatedly, it would be good to make a quantitative connection between the feedback loops proposed in Fig. 5 and the main results in Fig. 3. Is it possible to predict some of the qualitative behaviors from the feedbacks resulting from simulations? Fig. 2 also has potential implications for Fig. 3 that could be drawn out more. For example, higher interspecific competition tends to reduce coexistence (as expected) in both Fig. 2 and 3.

The analysis of the empirical data is lacking. The manuscript provides no details about how or what models are fit. More importantly, there is not any model comparison. For example, does a quadratic (or whatever hump shaped curve is in Fig. 6D) fit better than a line in Fig 6C? Reciprocally, does a line fit better than a quadratic in Fig. 6D? A robust statistical approach is required here and it needs to be described clearly. Similarly, how are the connectances being modified in the data? Could they be changing with other important ecological parameters?

The bifurcation diagrams in the Supplement are very beautiful. Are there other situations in which adaptive foraging produces these sorts of bifurcations?

READABILITY COMMENTS

Including the Methods section before the Results would greatly improve the reader’s ability to keep track of what is being simulated. As it is currently written, the reader has to skip ahead to know even the model form.

In Equation (2), why does the strength of the benefit to the exploiter depend on the ratio of the mutualist to the exploiter? The biological explanation of this modeling choice is unclear. Presumably, it is describing the interaction between the mutualist, exploiter and resource, but I am not sure why the ratio is the correct term for this, rather than simply the mutualist abundance. Do other functional forms for this mutualism drastically change the results?

When varying the connectance in Fig. 3, exactly which links are being turned on and off? From the Methods section, it was unclear to me whether or not connectance was modifying any link in the food web, only those involving the top predator or only those involving the exploiters. Clearly, some of these functionally interactions are not equivalent to the others, complicating the definition of connectance.

The Methods section would benefit from more discussion of the differential equations for theta. Is this functional form the same as other studies. It would be useful to point out that the equations constrain the sum of all theta values is constant throughout the dynamics.

Using the same color bar across panels in Fig. 2 would improve its clarity so a reader can see right away that some cases are always unstable.

**Have the authors made all data and (if applicable) computational code underlying the findings in their manuscript fully available?**

Reviewer #1: Yes

Reviewer #2: Yes

Reviewer #3: Yes

PLOS authors have the option to publish the peer review history of their article (what does this mean? ). If published, this will include your full peer review and any attached files.

**Do you want your identity to be public for this peer review?** For information about this choice, including consent withdrawal, please see our Privacy Policy .

Reviewer #1: No

Reviewer #2: No

Reviewer #3: No

**Figure resubmission:**
---

## [Editor Report · Decision Letter 1]

PCOMPBIOL-D-24-01756R1

Evolution of foraging behaviour induces variable complexity-stability relationships in mutualist-exploiter-predator communities

PLOS Computational Biology

Dear Dr. Wang,

Thank you for submitting your manuscript to PLOS Computational Biology. After careful consideration, we feel that it has merit but does not fully meet PLOS Computational Biology's publication criteria as it currently stands. Therefore, we invite you to submit a revised version of the manuscript that addresses the points raised during the review process.

Please submit your revised manuscript within 30 days Jul 21 2025 11:59PM. If you will need more time than this to complete your revisions, please reply to this message or contact the journal office at ploscompbiol@plos.org. Please include the following items when submitting your revised manuscript:

We look forward to receiving your revised manuscript.

Kind regards,

Samraat Pawar, PhD

Academic Editor

PLOS Computational Biology

Tobias Bollenbach

Section Editor

PLOS Computational Biology

**Additional Editor Comments :**

Thank you for submitting the revised manuscript.

We appreciate the care you have taken to address the previous reviews.

After evaluating the revision, I think a moderate further revision is necessary before the manuscript can be accepted:

* Appendix B varies one parameter at a time. Please add a multivariate exploration (e.g. Latin-hyper-cube sampling with partial-rank correlations) across all free parameters to demonstrate that the main complexity–stability patterns are robust to joint uncertainty. A concise summary figure or table would be sufficient.

* The qualitative match between model predictions and published food-web patterns in Fig 5 is still weak validation at best - please revise the discussion to acknowledge this more directly with suitable caveats, or provide a stronger validation (may not be possible).

* The manuscript still exceeds 9,000 words and contains several minor grammatical slips (e.g. “biomss”, “responds” instead of “responses”).

**Figure resubmission:**
---

## [Editor Report · Decision Letter 2]

Dear Dr. Wang,

We are pleased to inform you that your manuscript 'Evolution of foraging behaviour induces variable complexity-stability relationships in mutualist-exploiter-predator communities' has been provisionally accepted for publication in PLOS Computational Biology.

Best regards,

Samraat Pawar, PhD

Academic Editor

PLOS Computational Biology

Tobias Bollenbach

Section Editor

PLOS Computational Biology

Thank you for these final revisions.

I am happy to recommend acceptance.

---

## [Editor Report · Acceptance letter]

PCOMPBIOL-D-24-01756R2

Evolution of foraging behaviour induces variable complexity-stability relationships in mutualist-exploiter-predator communities

Dear Dr Wang,

I am pleased to inform you that your manuscript has been formally accepted for publication in PLOS Computational Biology. Your manuscript is now with our production department and you will be notified of the publication date in due course.

With kind regards,

Lilla Horvath
